# The Role of Macrophage Inhibitory Factor in TAA-Induced Liver Fibrosis in Mice: Modulatory Effects of Betaine

**DOI:** 10.3390/biomedicines12061337

**Published:** 2024-06-17

**Authors:** Tatjana Radosavljevic, Dusan Vukicevic, Jasmina Djuretić, Kristina Gopcevic, Milica Labudovic Borovic, Sanja Stankovic, Janko Samardzic, Milica Radosavljevic, Danijela Vucevic, Vladimir Jakovljevic

**Affiliations:** 1Institute of Pathophysiology “Ljubodrag Buba Mihailović”, Faculty of Medicine, University of Belgrade, 11000 Belgrade, Serbia; danijela.vucevic@med.bg.ac.rs; 2Uniklinik Mannheim, Theodor-Kutyer-Ufer 1-3, 68167 Mannheim, Germany; vukdusan@hotmail.com; 3Department of Pathobiology, Faculty of Pharmacy, University of Belgrade, 11000 Belgrade, Serbia; jasmina.djuretic@pharmacy.bg.ac.rs; 4Institute of Chemistry in Medicine “Prof. Dr. Petar Matavulj”, Faculty of Medicine, University of Belgrade, 11000 Belgrade, Serbia; kristina.gopcevic@med.bg.ac.rs; 5Institute of Histology and Embryology, Faculty of Medicine, University of Belgrade, 11000 Belgrade, Serbia; milica.labudovic-borovic@med.bg.ac.rs; 6Centre for Medical Biochemistry, University Clinical Centre of Serbia, 11000 Belgrade, Serbia; sanjast2013@gmail.com; 7Department of Physiology, Faculty of Medical Sciences, University of Kragujevac, Svetozara Markovica 69, 34000 Kragujevac, Serbia; drvladakgbg@gmail.com; 8Institute of Pharmacology, Clinical Pharmacology and Toxicology, Faculty of Medicine, University of Belgrade, 11000 Belgrade, Serbia; jankomedico@yahoo.es (J.S.); milica.radosavljevic.bg@gmail.com (M.R.); 9Center of Excellence for the Study of Redox Balance in Cardiovascular and Metabolic Disorders, University of Kragujevac, Svetozara Markovica 69, 34000 Kragujevac, Serbia; 10Department of Human Pathology, First Moscow State Medical University I.M. Sechenov, Trubetskaya Street 8, Str. 2, 119991 Moscow, Russia

**Keywords:** liver fibrosis, thioacetamide, macrophage migration inhibitory factor, betaine, mice

## Abstract

Macrophage inhibitory factor (MIF) is a multipotent cytokine, involved in the inflammatory response to infections or injuries. This study investigates the role of MIF in liver fibrosis and the modulating effect of betaine on MIF in thioacetamide (TAA)-induced liver fibrosis. The wild-type and knockout MIF−/− C57BL/6 mice were divided into the following groups: control; Bet group, which received betaine; MIF−/−; MIF−/−+Bet; TAA group, which received TAA; TAA+Bet; MIF−/−+TAA; and MIF−/−+TAA+Bet group. After eight weeks of treatment, liver tissue was collected for further analysis. The results revealed that TAA-treated MIF-deficient mice had elevated levels of hepatic TGF-β1 and PDGF-BB, as well as MMP-2, MMP-9, and TIMP-1 compared to TAA-treated wild-type mice. However, the administration of betaine to TAA-treated MIF-deficient mice reduced hepatic TGF-β1 and PDGF-BB levels and also the relative activities of MMP-2, MMP-9 and TIMP-1, albeit less effectively than in TAA-treated mice without MIF deficiency. Furthermore, the antifibrogenic effect of MIF was demonstrated by an increase in MMP2/TIMP1 and MMP9/TIMP1 ratios. The changes in the hepatic levels of fibrogenic factors were confirmed by a histological examination of liver tissue. Overall, the dual nature of MIF highlights its involvement in the progression of liver fibrosis. Its prooxidant and proinflammatory effects may exacerbate tissue damage and inflammation initially, but its antifibrogenic activity suggests a potential protective role against fibrosis development. The study showed that betaine modulates the antifibrogenic effects of MIF in TAA-induced liver fibrosis, by decreasing TGF-β1, PDGF-BB, MMP-2, MMP-9, TIMP-1, and the deposition of ECM (Coll1 and Coll3) in the liver.

## 1. Introduction

Liver fibrosis is a complex, potentially reversible process [1,2,3] involving multiple cellular and molecular mechanisms [4,5,6,7]. These fibrotic changes occur when there is an imbalance between the processes of extracellular matrix (ECM) formation and deposition, i.e., fibrogenesis, and the processes of matrix degradation and removal, i.e., fibrinolysis [2,8].

Organ fibrosis accounts for up to 45% of all-cause mortality worldwide and thus represents a significant unmet medical need [7,9]. Hepatic stellate cells (HSCs) play a crucial role in liver fibrogenesis. In a healthy liver, HSCs serve as storage for retinyl esters and contribute to liver regeneration [3]. However, in chronic liver disease, HSCs are activated, and contribute to the excessive production of fibrogenic and inflammatory molecules, leading to liver fibrosis [6,7,9]. Various factors influence the activation of HSCs. Oxidative stress, caused by the production of reactive oxygen species (ROS) and a decrease in antioxidant capacity, can directly or indirectly stimulate HSC activation. Proinflammatory cytokines, chemokines, and other pathological factors released by injured hepatocytes and activated Kupffer cells can also trigger HSC activation [6,7]. Two key profibrogenic cytokines implicated in liver fibrogenesis are transforming growth factor-beta 1 (TGF-β1) and platelet-derived growth factor-BB (PDGF-BB). TGF-β1 stimulates the transdifferentiation of HSCs into myofibroblasts, which are accountable for synthesizing and releasing collagen and other ECM components, such as fibronectin. Fibronectin plays a pivotal role in cell growth, migration, adhesion, and differentiation [10]. PDGF-BB serves as a potent mitogen for HSCs, fostering their proliferation, chemotaxis, migration, transdifferentiation, and collagen synthesis and deposition within the ECM [11]. 

In addition, matrix metalloproteinases (MMPs) play an important role in the ECM remodeling processes associated with liver fibrogenesis. An imbalance between MMPs and tissue inhibitors of metalloproteinases (TIMPs) can lead to the excessive deposition of ECM and tissue remodeling, thereby exacerbating the progression of liver fibrosis [12]. MMP2 and MMP9 are crucial MMPs produced by HSCs. Their activities are tightly regulated by TIMPs, which ensure a balance between ECM degradation and synthesis. MMP-2 and MMP-9 are expressed in the early stages of liver damage. In the later stages, the inhibition of fibrillar collagen degradation exceeds ECM synthesis. This is evidenced by the increased expression of TIMP-1 in the liver, which protects HSCs from apoptosis and reduces MMP-mediated collagen degradation [12]. The determination of the ratio of MMPs to TIMPs is a promising marker for the assessment of liver fibrosis. This ratio is a key indicator of the balance between extracellular matrix (ECM) degradation and synthesis, which has a direct impact on the progression of fibrotic processes in the liver [13].

Macrophage migration inhibitory factor (MIF) is a multifaceted cytokine implicated in the progression of various chronic inflammatory disorders [14,15,16,17,18,19,20]. In addition to immune cells, MIF can also be produced by non-immune cells such as hepatocytes, endothelial cells, epithelial cells, and tumor cells [14,21]. Elevated levels of MIF expression have been observed in hepatocytes from patients with conditions such as alcoholic liver disease (ALD), non-alcoholic fatty liver disease (NAFLD), liver cirrhosis, and hepatocellular carcinoma (HCC) [22,23,24]. In liver fibrosis, MIF plays a role in chronic inflammation and fibrogenesis, exhibiting pro-inflammatory and anti-inflammatory properties depending on the etiology and stage of liver disease [22]. MIF promotes inflammation by stimulating the release of cytokines such as interleukin (IL)-6, IL-1β, and interferon-gamma (IFN-γ) from the immune cells that activate HSCs and hepatocytes [25]. Activated HSCs produce cytokines and chemokines that sustain chronic inflammation and fibrosis progression. Hepatocytes, responsible for metabolizing toxins, can be damaged in liver injury, leading to oxidative stress and the release of pro-inflammatory and fibrogenic cytokines. Kupffer cells, specialized liver macrophages, are sensitive to ROS and pro-inflammatory cytokines and contribute to oxidative stress, inflammation, and fibrogenesis by releasing various factors, including MIF [24]. In addition, MIF is upregulated in the fibrotic liver tissue of rats treated with thioacetamide (TAA), a hepatotoxin commonly used to induce liver fibrosis [26]. However, there is conflicting evidence that MIF may have antifibrotic effects in the experimental models of hepatotoxin-induced liver fibrosis by reducing myofibroblast activity rather than altering immune cell infiltration [27].

TAA-induced hepatotoxicity is commonly used as an animal model to study acute liver injury, fibrosis, and cirrhosis [28,29]. TAA undergoes liver bioactivation, leading to the formation of reactive metabolites, particularly its SS-dioxide, which are thought to cause liver injury [30]. TAA-induced liver fibrosis is associated with oxidative stress, inflammation, impaired fatty acid metabolism, and lipid peroxidation, making antioxidants an area of great interest for prevention and treatment [31]. MIF has oxidoreductase activity and increases cellular glutathione levels, which is an important antioxidant. By reducing hepatocyte death, a significant contributor to liver injury, MIF may exert hepatoprotective effects and potentially hinder fibrosis progression [32]. However, the exact role of MIF in liver fibrosis and the underlying mechanisms remain unclear.

Betaine, also known as trimethylglycine, is a naturally occurring compound found in plants, animals, and microorganisms, and is abundant in various dietary sources such as wheat germ, bran, vegetables, and seafood [33]. It is an oxidative metabolite of choline and exhibits antioxidant properties by enhancing S-adenosylmethionine (SAM) and methionine, important molecules involved in antioxidant metabolic pathways. Betaine supplementation has been shown to have the potential to attenuate oxidative stress and its consequences in ALD and NAFLD, as well as reduce fibrotic and necrotic liver lesions [6,34,35,36]. Nevertheless, the exact antifibrotic effects of betaine in TAA-induced liver fibrosis development and the underlying mechanisms remain incompletely understood.

Gaining insight into the underlying mechanisms of liver fibrosis is essential for the development of novel therapeutic approaches. The aim of this study was to investigate the role of MIF in liver fibrogenesis. Given the importance of antioxidants in the prevention and treatment of chronic liver diseases, this study also examines the effect of betaine supplementation on MIF in TAA-induced liver fibrosis. This study is the first to demonstrate the modulating effect of betaine on MIF-mediated liver fibrosis. Therefore, the results of this research will provide important insights into liver fibrogenesis and its potential therapeutic approach. Such an investigation should improve our understanding of the therapeutic efficacy of betaine and its mechanisms of action in the treatment of liver fibrosis.

## 2. Materials and Methods

### 2.1. Animals

The experiment was performed on wild-type C57BL/6 male mice and MIF knockout C57BL/6 mice (MIF−/−) 8 weeks old, weighing 21–25 g, raised at the Military Medical Academy, Belgrade. Animals were kept under standard laboratory conditions (temperature 22 ± 2 °C, relative humidity 50 ± 10%, 12/12 light-dark cycle with lights turned on at 9.00 a.m.). The animals were kept in groups of six mice per cage with free access to tap water and a standard chow diet for laboratory mice. All experimental procedures were in full compliance with the Directive of the European Parliament and the Council (2010/63EU) and approved by the Ethical Committee of the University of Belgrade (Permission No 6600/2).

### 2.2. Setting an Animal Model of Liver Fibrosis

Before the experiment, all animals (n = 80) were fed with a control diet. At the age of 8 weeks, they were divided into several groups. The first group, the control group (C), had free access to tap water and a standard chow diet (n = 10). The second group, the Bet group, was fed with a standard chow diet and supplemented with betaine (n = 10). The third group, the MIF−/− group, consisted of MIF−/− mice continuously fed with a standard chow diet (n = 10). The fourth group, the MIF−/−+Bet group, consisted of MIF−/− mice fed with a standard chow diet and supplemented with betaine (n = 10). The fifth group, the TAA group, was fed with a standard chow diet and treated with thioacetamide (TAA) (n = 10). The sixth group, the TAA+Bet group, consisted of wild-type mice treated with TAA and supplemented with betaine (n = 10). The seventh group, the MIF−/−+TAA group, consisted of MIF−/− mice continuously fed with a standard chow diet and treated with TAA (n = 10). The eighth group, the MIF−/−+TAA+Bet group, consisted of MIF−/− mice continuously fed with a standard chow diet, treated with TAA, and supplemented with betaine (MIF; n = 10). Chronic liver inflammation was induced by TAA (200 mg/kg) dissolved in 200 μL PBS, intraperitoneally, three times a week during an 8-week period. The working solution was stored at 4 °C. The fresh solution was prepared every week. Simultaneously, the C, Bet, MIF−/−, and MIF−/−+Bet groups received the vehicle (saline, 0.9% NaCl) in the same manner. Betaine (MP Biomedicals) was dissolved in drinking water (2% wt/v), and animals had free access to drink *ad libitum*. The mice were fasted overnight the day before the sacrifice. After the treatment, they were sacrificed by exsanguination under ketamine (100 mg/kg intraperitoneally/i.p./) anesthesia. Liver samples were collected for examination for fibrogenic mediators and histopathological analysis.

### 2.3. Determination of the Liver Profibrogenic Mediators (TGF-β1 and PDGF-BB) 

For the determination of TGF-β1 and PDGF-BB in liver tissue, samples were homogenized in 10 volumes of PBS. After centrifugation (10 min at 12,000× *g*, 4 °C), the supernatants were carefully collected through the fat cake and diluted to 1/40,000 in PBS. To determine the concentration of TGF-β1 and PDGF-BB in liver tissue, we applied ELISA kits from ELABSCIENCE (Houston, TX, USA), according to the manufacturer’s instructions.

### 2.4. Determination of Liver MMP-2, MMP-9, Dimer MMP-9 and TIMP-1 by Zymography 

The activities of mouse gelatinase A (mouse MMP-2, P33434), gelatinase B (mouse MMP-9, P41245), and TIMP-1 were determined by gelatin zymography [37,38]. For each sample, an equal total tissue homogenate protein concentration was loaded after protein concentration determination. The tissue homogenates were diluted with 20% sucrose solution (10 μL homogenate and 1190 μL sucrose solution, *v*/*v*) and incubated in a thermostatically controlled water bath at 37 °C for 45 min. After incubation, homogenates were mixed with 2 × zymography sample buffer [0.125 M Tris–HCl, pH 6.8, 20% (*v*/*v*) glycerol, 4% (*w/v*) SDS, and 0.005% bromophenol blue], and then loaded into SDSPAGE that was performed on 8% acrylamide gels containing 0.1% (*w*/*v*) gelatin (Sigma–Aldrich, St. Louis, MO, USA). The negative control consisted of tissue homogenates incubated with 5 mmol/L EDTA. Human recombinant gelatinases A and B (R&D Systems, Minneapolis, MN, USA) were employed as standard. The negative control contained a mixture of standard human recombinant gelatinases A and B and 5 mmol/L EDTA. After electrophoresis, the gel was washed twice for 30 min in zymography renaturing buffer (2.5% Triton X-100) with a gentle shake to remove SDS, then incubated for 24 h at 37 °C in reaction buffer (50 mM Tris–HCl, pH 7.4, 200 mM NaCl, and 5 mM CaCl_2_). The gels were stained for 2 h with a Coomassie brilliant blue before being destained with a destaining solution (50% methanol, 10% acetic acid, and 40% ddH_2_O). Finally, the gelatinolytic activities on the gels were presented as transparent bands on the blue background. The gelatinolytic activities were identified as clear zones. The densitometric value of the lyses against a dark blue background on zymography gels was calculated using the ImageJ 1.42q software package (National Institutes of Health, Bethesda, MD, USA), which quantified both the surface and the intensity of the lysis bands after scanning the gels. The relative activities of gelatinases A and B were expressed as percentages of total activity, which was taken as 100%. The relative activities of the gelatinases were compared using Student’s *t*-test.

### 2.5. Histology Analysis of Liver Tissue

Liver tissue was sectioned and incubated in a 10% formalin solution at room temperature. After fixation, the liver samples were processed by the standard method. Tissues were incorporated in paraffin, sectioned at 5 μm, and then stained with Hematoxylin-Eosin (HE), Masson’s trichrome, and reticulin, according to the manufacturer’s instructions. The sections were analyzed and photographed using an Olympus BX51 (Olympus, Tokyo, Japan) light microscope equipped with an Artcore 500 MI (Artray, Co., Ltd., Tokyo, Japan) camera. Fibrosis was quantified by analyzing liver tissue samples stained with Masson’s trichrome and Reticulin with a digital image camera. The surface of the blue-stained area at a microscope magnification of ×200 was calculated in 10 random fields on each section of each animal and presented as a percentage of the total liver cross-sectional area using the ImageJ software.

### 2.6. Statistical Analysis 

All results are expressed as means ± SD. As the normal distribution of parameters was confirmed by the Kolmogorov–Smirnov test, and a one-way analysis of variance (ANOVA) with Tukey’s post hoc test was used to test the difference among groups. Statistical significance was set at *p* < 0.05. Computer software SPSS 15.0 was used for the statistical analysis.

## 3. Results

### 3.1. The Effects of MIF and Betaine on Profibrogenic Mediators (TGF-β1 and PDGF-BB) in TAA-Induced Liver Fibrosis

Our results have shown that liver TGF-β1 concentration was significantly higher in the TAA group (170.12 ± 24.89 pg/mL) in comparison with the control values (85.60 ± 10.92 pg/mL) (*p* < 0.01). Liver TGF-β1 concentration was significantly higher in MIF−/−+TAA (230.49 ± 34.45 pg/mL) and significantly decreased in the TAA+Bet group (126.16 ± 20.76 pg/mL) compared to the TAA group (*p* < 0.01). Moreover, the level of TGF-β1 in the liver was significantly decreased in the MIF−/−+TAA+Bet group (172.15 ± 26.25 pg/mL) compared to the MIF−/−+TAA group (*p* < 0.01) and significantly higher compared to the TAA+Bet group (*p* < 0.01) (Figure 1A). Similarly, the results of our study have shown that liver PDGF-BB concentration was significantly higher (*p* < 0.01) in the TAA group (2199.86 ± 322.63 pg/mL) in comparison with the control values (760.53 ± 101.41 pg/mL). Liver PDGF-BB concentration was significantly higher in the MIF−/−+TAA (2452.72 ± 129.63 pg/mL) and significantly decreased in the TAA+Bet group (1135.65 ± 144.49 pg/mL) compared to TAA group (*p* < 0.05; *p* < 0.01, respectively). Moreover, the level of PDGF-BB in the liver was significantly decreased in the MIF−/−+TAA+Bet group (1859.66 ± 347.43 pg/mL) compared to the MIF−/−+TAA group (*p* < 0.01) and significantly higher compared to the TAA+Bet group (*p* < 0.01) (Figure 1B). For TGF-β1 and PDGF-BB parameters, there was no significance in the Bet, MIF−/−, and MIF−/−+Bet groups compared to the control group (*p* > 0.05, respectively) (Figure 1). 

Our results demonstrated the antifibrogenic effect of MIF (decreased TGF-β1 and PDGF-BB levels) in TAA-induced liver fibrosis, together with the protective effect of betaine. In particular, betaine enhanced the antifibrogenic effect of MIF.

Abbreviations: C, control group; MIF−/− group, mice knockout for macrophage migration inhibitory factor (MIF); Bet, betaine group; MIF−/−+Bet group, knockout MIF mice who have received betaine; TAA group, animals who have received thioacetamide (TAA); MIF−/−+TAA group, knockout MIF mice who have received thioacetamide; TAA+Bet group, animals who have received thioacetamide and betaine; MIF−/−+TAA+Bet group, knockout MIF mice who have received thioacetamide and betaine.

### 3.2. Effects of MIF and Betaine on MMP-2, MMP-9, Dimer MMP-9 and TIMP-1 Activity in TAA-Induced Liver Fibrosis

Gelatin zymography revealed that tissue homogenates contained both MMP-2 (72 kDa) and MMP-9 (92 kDa). MMP-9 also appeared in a dimer form (~220 kDa) in all liver tissue samples (Appendix A). Densitograms of the gelatin zymography of MMPs in the tissue homogenates of TAA-induced liver fibrosis in mice are presented in Appendix A. The densitometric area (Arbitrary Units) of MMP-2, MMP-9, dimer MMP-9, and TIMP-1 in liver tissue homogenates were significantly increased in the TAA group in comparison with the control values (*p* < 0.01) (Figure 2). The densitometric area of MMP-2, MMP-9, and TIMP-1 was significantly increased in the MIF−/−+TAA and significantly decreased in the TAA+Bet group compared to the TAA group (*p* < 0.05; *p* < 0.01) while the densitometric area of dimer MMP-9 was significantly decreased in MIF−/−+TAA compared to TAA+Bet and TAA group (*p* < 0.01). Moreover, MMP-2, MMP-9, and TIMP were decreased considerably in the MIF−/−+TAA+Bet group compared to the MIF−/−+TAA group (*p* < 0.01) and significantly increased compared to the TAA+Bet group (*p* < 0.01) (Figure 2). However, the densitometric area of dimer MMP-9 in the MIF−/−+TAA+Bet group was significantly decreased compared to the TAA+Bet group (*p* < 0.01) (Figure 2). 

This study showed the antifibrogenic effect of MIF (increased activity of MMP-2, MMP-9 and reduced levels of TIMP1) in TAA-induced liver fibrosis. Notably, betaine stimulated the antifibrogenic effect of MIF.

### 3.3. The Effects of MIF and Betaine on the Imbalance of MMP2/TIMP1 and MMP9/TIMP1 in the Livers of Mice with TAA-Induced Liver Fibrosis

Our results have shown that MMP2/TIMP1 ratio was increased in the TAA group in comparison with the control values (*p* < 0.05). This ratio was significantly higher in TAA+Bet group and significantly decreased in the MIF−/−+TAA group compared to TAA group (*p* < 0.01). Moreover, the ratio of MMP2/TIMP1 was increased in the MIF−/−+TAA+Bet group compared to MIF−/−+TAA group (*p* < 0.01) (Figure 3). Similarly, the results of our study have shown that MMP9/TIMP1 ratio was significantly higher in the TAA group in comparison with the control values (*p* < 0.05). This ratio was reduced in the MIF−/−+TAA and increased in the TAA+Bet group compared to TAA group (*p* < 0.01). Moreover, the ratio of MMP9/TIMP1 was increased in the MIF−/−+TAA+Bet group compared to the MIF−/−+TAA group and decreased compared to the TAA+Bet group (*p* < 0.01; *p* < 0.05, respectively) (Figure 3). For the ratios MMP2/TIMP1 and MMP9/TIMP1, there were no significance in MIF−/−, and MIF−/−+Bet group compared to the control group (Figure 3).

### 3.4. Effects of MIF and Betaine on Histology Findings of Liver Morphology in Mice with TAA-Induced Liver Fibrosis

The histopathology findings of liver tissue in TAA-treated wild type animals show the irregular structure of the liver with mononuclear inflammatory infiltrate in the portal and perivascular spaces. Mild liver fibrosis and the microvesicular fatty change was present. Moreover, numerous hepatocytes were with bizarre, polyploid nuclei that contained multiple nucleoli. Mallory bodies and hemosiderophages were also observed. Rare focal hepatocyte necrosis was present. The number of mitoses was 0/10 HPF to 1/10HPF (High Power Field) (Figure 4a).

Betaine supplementation of TAA-treated wild-type animals improved the histological findings of liver tissue (Figure 4b). The histological structure of the liver was less irregular, with scanty fibrosis (bridging porto-portal) and mild inflammatory infiltrates. Mallory bodies, rare hemosiderophages, and rare focal hepatocyte necrosis were observed.

MIF knockout mice treated with TAA had a more pronounced irregular liver structure with a mixed inflammatory infiltrate in the Disse spaces and perivascularly along large blood vessels compared to wild-type C57BL/6 mice treated with TAA (Figure 4c). Pronounced fibrosis (bridging porto-portal) and micronodular cirrhosis were also present. Moreover, micro- and macrovesicular fatty changes were present. Focal necrosis and apoptosis of hepatocytes were present. The number of mitoses was 0/10 HPF to 4/10 HPF (Figure 4c). 

After betaine supplementation, MIF knockout C57BL/6 mice had the irregular histological structure of liver tissue with mild fibrosis (bridging porto-portal) and mixed inflammatory infiltrates. Micro- and macrovesicular fatty changes were present. Mallory bodies and rare apoptotic bodies were observed. Kupffer cell hyperplasia and rare focal necrosis of hepatocytes were present (Figure 4d).

### 3.5. The Effects of MIF and Betaine on TAA-Induced Liver Fibrosis in Mice

TAA-induced liver fibrosis in mice was characterized by the accumulation of ECM (type I and III collagen), which was detected with Masson’s trichrome (Figure 5) and reticulin staining (Figure 6). The quantification of liver fibrosis in the TAA group (2.15 ± 0.20%) showed a significantly increased accumulation of ECM (type I collagen) compared to control (0.15 ± 0.02%) (*p* < 0.01). However, the fibrosis score was significantly higher in the liver of TAA-treated MIF deficient mice (5.45 ± 0.65%) compared to wild-type TAA-treated mice (*p* < 0.01). Betaine supplementation in TAA-treated mice mitigates liver fibrosis (1.45 ± 0.15%) in comparison with the TAA group *(p* < 0.05). Also, betaine supplementation in TAA-treated MIF-deficient mice reduced the accumulation of ECM (3.23 ± 0.41%) compared to the MIF−/−+TAA group, while it increased compared to the TAA+Bet group (*p* < 0.01). (Figure 5). Similar to this, the quantification of liver fibrosis in the TAA group (2.35 ± 0.21%) showed a significantly increased accumulation of ECM (type III collagen) compared to control (0.14 ± 0.03%) (*p* < 0.01). However, the fibrosis score was significantly higher in the liver of TAA-treated MIF deficient mice (5.15 ± 0.67%) compared to wild-type TAA-treated mice (*p* < 0.01). Betaine supplementation in TAA-treated mice mitigates liver fibrosis (1.25 ± 0.15%) in comparison with the TAA group (*p* < 0.05). Also, betaine supplementation in TAA-treated MIF deficient mice reduced the accumulation of ECM (3.72 ± 0.40%) compared to the MIF−/−+TAA group, while it was increased compared to the TAA+Bet group (*p* < 0.01) (Figure 6).

## 4. Discussion

Our study showed a significant increase in hepatic TGF-β1 and PDGF-BB levels in TAA-treated MIF-deficient mice compared to TAA-treated wild-type mice. These results suggest that MIF may possess antifibrogenic properties, possibly by inhibiting HSC activation, promoting ECM degradation, and facilitating fibrosis resolution [39,40]. MIF exerts its antifibrogenic effect on HSCs via the CD74 receptor [39]. In addition, MIF inhibits the PDGF-induced migration and proliferation of HSCs, resulting in the reduced deposition of ECM and the progression of liver fibrosis. The MIF-induced CD74/AMPK signaling pathway also suppresses TGF-β-induced Coll1 expression [41,42]. Therefore, the enhanced fibrogenesis observed in MIF-deficient mice is likely due to the role of MIF in HSC biology rather than altered immune cell infiltration [39]. Further studies are needed to fully elucidate the effects of MIF via CD74 receptors on HSCs. A recent study has shown that the MIF/CD74 signaling pathway, which promotes autophagy in HSCs, may contribute significantly to the pathogenesis of liver fibrosis [43]. In a rat model of TAA-induced fibrosis, damaged hepatocytes adjacent to fibrotic areas were identified as a source of MIF [44]. While elevated MIF levels are typically associated with inflammation in most liver diseases [45], the exact effect on liver fibrogenesis remains ambiguous. The proinflammatory and fibrogenic effect of MIF has been demonstrated in experimental models of ALD, as well as in the human population, and correlates with patient mortality in alcoholic hepatitis [42,46,47]. In ALD, hepatocytes produce MIF that interacts with the CD74 receptor and its co-receptors CXCR2, CXCR4, and CXCR7, which are expressed on resident hepatic macrophages and peripheral monocytes [40,48]. Studies have shown that liver and blood MIF levels are elevated in acute CCl4 toxicity and CCl4-induced liver fibrosis, indicating the local and systemic effects of MIF in the progression of chronic liver disease [24,49,50]. However, MIF exhibits protective effects following chronic binge ethanol feeding [24] and exerts an anti-steatotic effect in methionine-choline-deficient (MCD) diet-induced NAFLD [51]. At the same time, it promotes hepatic fibrogenesis in MCD diet-induced NASH [52]. Furthermore, MIF contributes to liver fibrogenesis in NAFLD/NASH progression through the profibrotic phenotype of NKT cells [53] but attenuates liver fibrosis in models of toxic liver injury [40]. The role of MIF in liver disease varies based on the disease etiology, stage, specific pattern of intrahepatic MIF receptor expression, and complexity of the MIF signal [53,54]. Recent research suggests that high serum levels of MIF and low levels of its CD74-binding receptor in the blood indicate an increased risk of mortality in patients with advanced liver cirrhosis [45]. Further exploration of the role of MIF in both liver and neurodegenerative diseases is crucial to identify MIF as a diagnostic and therapeutic target.

In this study, TAA-induced liver fibrosis was associated with a significant increase in MMP2 and MMP9 activity in the liver. Our findings indicate that the antifibrogenic effect of MIF is manifested by increasing the MMP2/TIMP1 and MMP9/TIMP1 ratios. Similarly, an overexpression of these enzymes and altered MMP2/TIMP1 and MMP9/TIMP1 ratios were also observed in the study by Ren et al. [13]. The regulation of MMPs/TIMPs is crucial for ECM degradation and remodeling, especially in liver fibrosis [55]. The changes in both MMP or TIMP level may modify the MMP/TIMP ratio, leading to the degradation or accumulation of the ECM [13]. MMP-1 and MMP-9 exhibit a negative correlation with histopathological findings in liver fibrosis [55]. MMP-2 activity is significantly increased in patients with liver fibrosis compared to controls and serves as a diagnostic marker [56]. As liver fibrosis progresses, serum TIMP levels increase, leading to an interaction of TIMP-1 with MMP-1, MMP-2, and MMP-9 [57]. In our study, increased levels of MMP-2, MMP-9, and TIMP-1 were observed in TAA-treated MIF-deficient mice compared to TAA-treated wild-type mice after a period of 8 weeks. MIF stimulates the secretion of MMPs [58,59], which explains the increased extent of fibrosis in MIF-deficient mice. TGF-β1 and PDGF-BB are important profibrogenic mediators that stimulate the synthesis of Coll1 and Coll3, laminin, fibronectin, and α-SMA [60]. Conversely, increased TIMP-1 expression and decreased MMP activity contribute to ECM remodeling [4,5]. TIMP-1 also has an anti-apoptotic effect on HSCs, possibly increasing their activation [4]. An increased expression of TIMP-1 has been observed in various liver fibrosis models [61]. Our results suggest that the antifibrogenic activity of MIF may be more pronounced through decreased TIMP activity than the activity of MMP-2. Furthermore, the enhanced fibrogenesis in MIF−/− mice contributes to the increased relative activity of MMP-9. On the other hand, the reduced relative activity and densitometric area of the MMP-9 dimer in the MIF−/−+TAA group compared to TAA-treated wild mice may be explained by the absence of the prooxidant and proinflammatory effects of MIF.

Betaine (trimethylglycine) exhibits antioxidant effects by regulating the metabolism of sulfur-containing amino acids (SAA), including homocysteine, methionine, and SAM [62]. It also acts as an osmoprotectant, shielding cells from stressors [63,64]. Various studies have highlighted the hepatoprotective effects of betaine in different experimental models [65,66,67,68]. However, this is the first study that highlighted the modulatory effects of betaine on MIF in TAA-induced liver fibrosis. Betaine supplementation significantly decreases hepatic TGF-β1 and PDGF-BB levels, as well as the relative activities of MMP-2, MMP-9, dimer MMP-9, and TIMP-1 in TAA-treated mice. Recent research suggests that TGF-β1 stimulates HSC activation by activating methionine adenosyltransferase 2A (MAT2A), leading to a reduction in SAM concentration [69]. This indicates a novel TGF-β1 signaling pathway in HSC activation and differentiation, pivotal in hepatic fibrogenesis [69]. As a regulator of SAA metabolism, betaine increases SAM synthesis [70], potentially inhibiting the TGF-β1/p65/MAT2A signaling pathway. Studies in various liver fibrosis models, including ALD, CCl4-induced fibrosis, high-fat diet induced fibrosis, and NASH, have shown that betaine treatment suppresses HSC activation, inflammation, and apoptosis while stimulating autophagy [65,67,68,71]. Our findings suggest that betaine reduces MMP-2, MMP-9, and TIMP-1 activities in liver fibrosis, with TIMP-1 reduction being the most prominent. It is known that ROS can activate MMPs by the oxidation or modification of amino acids [72]. The antioxidant properties of betaine may explain the decrease in MMP activity, as observed with silymarin in human melanoma cells [73]. In different experimental models, betaine also alleviated liver fibrosis [67,74,75]. Based on our previous results [76], it can be observed that the antifibrotic properties of betaine could be mediated through its antioxidative and anti-inflammatory effects, decreasing lipid peroxidation, protein oxidative damage, as well as IL-6 and IFN-γ levels. The antioxidative and anti-inflammatory effects of betaine contribute to its hepatoprotective properties in TAA-induced liver fibrosis [76]. The preservation of the antioxidant defense capacity by increasing thiol and decreasing TOS and MDA may explain the hepatoprotective activity of betaine [76]. Furthermore, betaine has been shown to regulate lipid metabolism and reduce the accumulation of triglycerides and cholesterol in ALD/NAFLD/NASH [68,75,77,78]. The more precise effects and mechanisms of betaine on suppressing hepatic fibrogenesis are still unclear. The administration of betaine to TAA-treated MIF-deficient mice results in an attenuated antifibrogenic response as measured by the liver levels of TGF-β1 and PDGF-BB and the relative activities of MMP-2, MMP-9, and TIMP-1. These results suggest that the absence of the antifibrogenic effect of MIF may contribute to this outcome. The antifibrogenic efficacy of betaine in MIF-deficient mice appears to be less pronounced compared to the TAA+ Bet group, suggesting that MIF exhibits antifibrogenic activity that is stimulated by betaine. However, betaine supplementation in MIF-deficient mice does not alter dimer MMP-9 activity, indicating that the antifibrogenic effect of MIF via dimer MMP-9 is independent of betaine. Moreover, our results indicate that betaine has antifibrogenic effect and stimulates MIF-mediated antifibrogenic activity through an increased MMP2/TIMP1 ratio. MIF exerts its antifibrogenic effect on HSCs via CD74 receptors, inhibiting their activation and PDGF-mediated proliferation [39]. The results of our study indicated that oxidative stress and inflammation were reduced in MIF-deficient mice treated with TAA, while liver fibrosis was exacerbated. MIF exhibits prooxidant and proinflammatory effects [76] in TAA-induced liver fibrosis but suppresses fibrosis, i.e., it has an antifibrogenic effect.

Chronic liver inflammation drives fibrosis, although the mechanisms governing fibrogenesis and inflammation differ. Th1 lymphocytes release cytokines like IFN-γ and TNF, activating M1 macrophages, which further perpetuate inflammation through cytokines and MMPs. Th2 lymphocyte-produced cytokines (IL-4 and IL-13) prompt the formation of anti-inflammatory M2 macrophages [79]. Liver macrophages, comprising Kupffer cells and bone marrow-derived monocytes, play a dual role. Kupffer cells stimulate inflammation and attract macrophages derived from monocytes. Monocyte-derived macrophages produce profibrogenic factors (TGF-β and PDGF) and contribute to HSC activation resembling M2 macrophages. However, they also aid in fibrinolysis by activating MMPs, facilitating ECM remodeling, fibrous tissue degradation, and the resolution of fibrosis [50,80]. 

Changes in the hepatic level of the fibrogenic factors (TGF-β1 and PDGF-BB), as well as in the activity of MMPs and TIMPs in the MIF−/−+TAA+Bet group, were confirmed by a histological examination of liver tissue. The pathohistological changes in liver tissue, as well as a higher percentage of fibrosis in the MIF−/−+TAA+Bet group compared to the TAA+Bet group, are probably due to the lack of MIF. Our previous results have shown that in TAA-induced hepatic fibrogenesis, MIF exerts prooxidant and proinflammatory effects [76], and this study has demonstrated the antifibrogenic effects of MIF. Similar to our results, Heinrichs et al. have shown that, in TAA and CCl4-induced liver fibrosis, MIF exerts antifibrogenic effects through CD74 receptors on HSC [39]. On the other hand, recent research indicates that MIF exhibits profibrogenic activity in NASH [53] and CCl4-induced liver fibrosis [43]. Our study showed that the antifibrogenic effects of betaine are MIF-mediated in TAA-induced liver fibrosis. These findings suggest that betaine could be used as a natural and non-toxic substance in the prevention of liver fibrosis and antioxidant therapy. Further research should focus on investigating how betaine can stimulate the MIF-mediated antifibrogenic response.

## 5. Conclusions

The results of this study implicate MIF as a possible target and betaine as a therapeutic factor for the prevention and treatment of MIF-mediated chronic liver diseases. The dual nature of MIF highlights its involvement in the progression of liver fibrosis. Its prooxidant and proinflammatory effects may exacerbate tissue damage and inflammation initially, but its antifibrogenic activity suggests a potential protective role against fibrosis development. Our study could contribute to a preventive and therapeutic approach to liver fibrosis. Further examinations and the clinical validation of these findings could improve therapeutic options for patients with liver fibrosis.

## Figures and Tables

**Figure 1 biomedicines-12-01337-f001:**
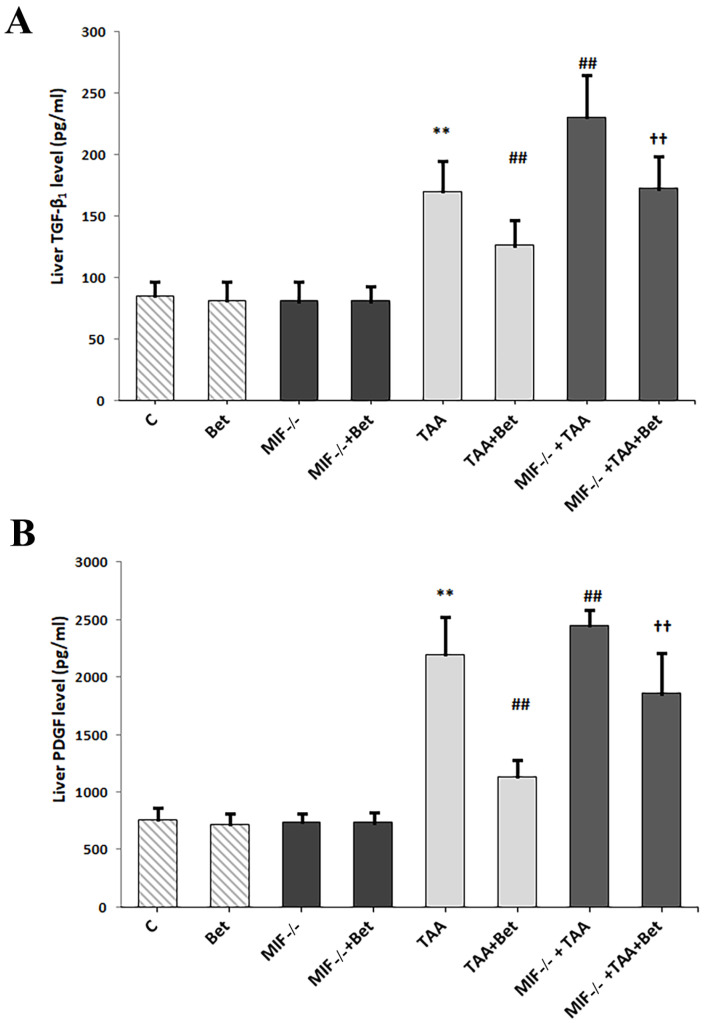
Effects of MIF and betaine on the liver level of TGF-β1 (**A**) and PDGF-BB (**B**) in mice with liver fibrosis. The data are presented as mean ± SD from 10 mice per group. Significance of the difference was estimated by using one-way analysis of variance (ANOVA) with Tukey’s post hoc test; ** *p* < 0.01 vs. C; ^##^
*p* < 0.01 vs. TAA; ^††^
*p* < 0.01 vs. MIF−/−+TAA and TAA+Bet.

**Figure 2 biomedicines-12-01337-f002:**
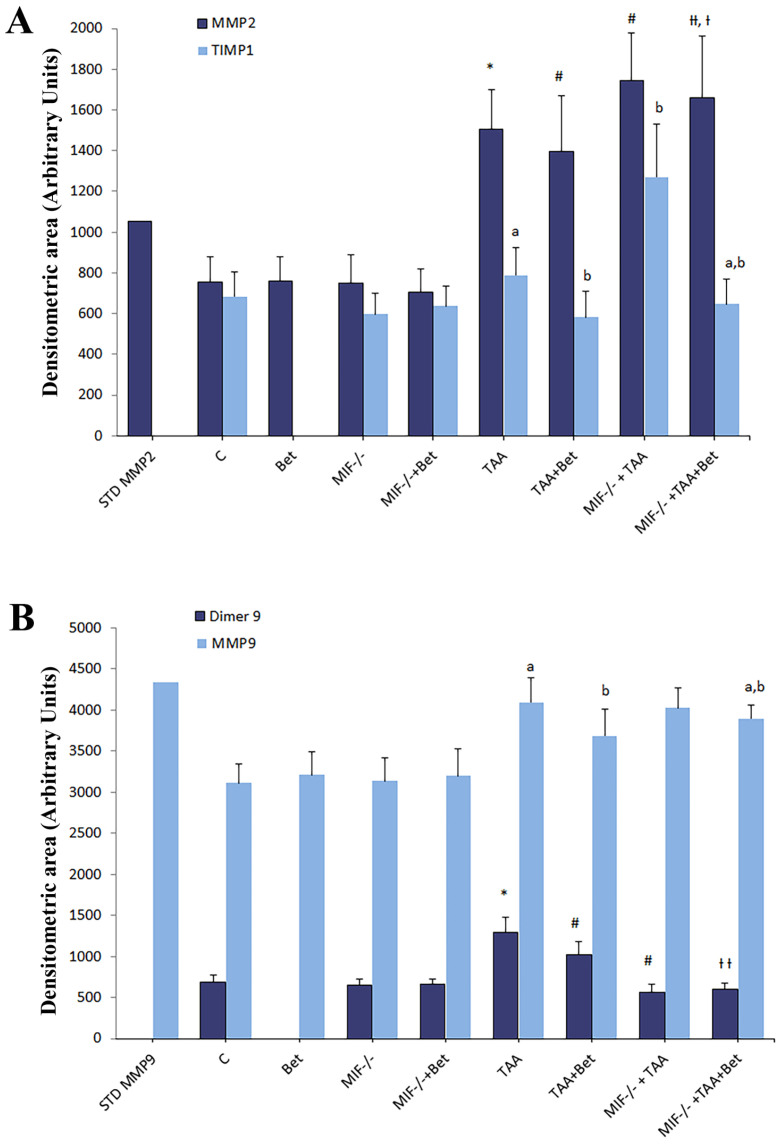
Effects of MIF and betaine on liver activity of MMP-2 and TIMP-1 (**A**), and dimer MMP-9 and MMP-9 (**B**), in TAA-induced liver fibrosis in mice. The data are presented as mean ± SD from 10 mice per group. Significance of the difference was estimated by using one-way analysis of variance (ANOVA) with Tukey’s post hoc test; * *p* < 0.01, ^a^
*p* < 0.01, vs. C; ^#^
*p* < 0.01, ^b^
*p* < 0.01, vs. TAA; ^††^
*p* < 0.01, ^a,b^
*p* < 0.01 vs. TAA+Bet; MIF−/−+TAA; ^†^
*p* < 0.05, ^a,b^
*p* < 0.05 vs. MIF−/−+TAA. For abbreviations, see Figure 1.

**Figure 3 biomedicines-12-01337-f003:**
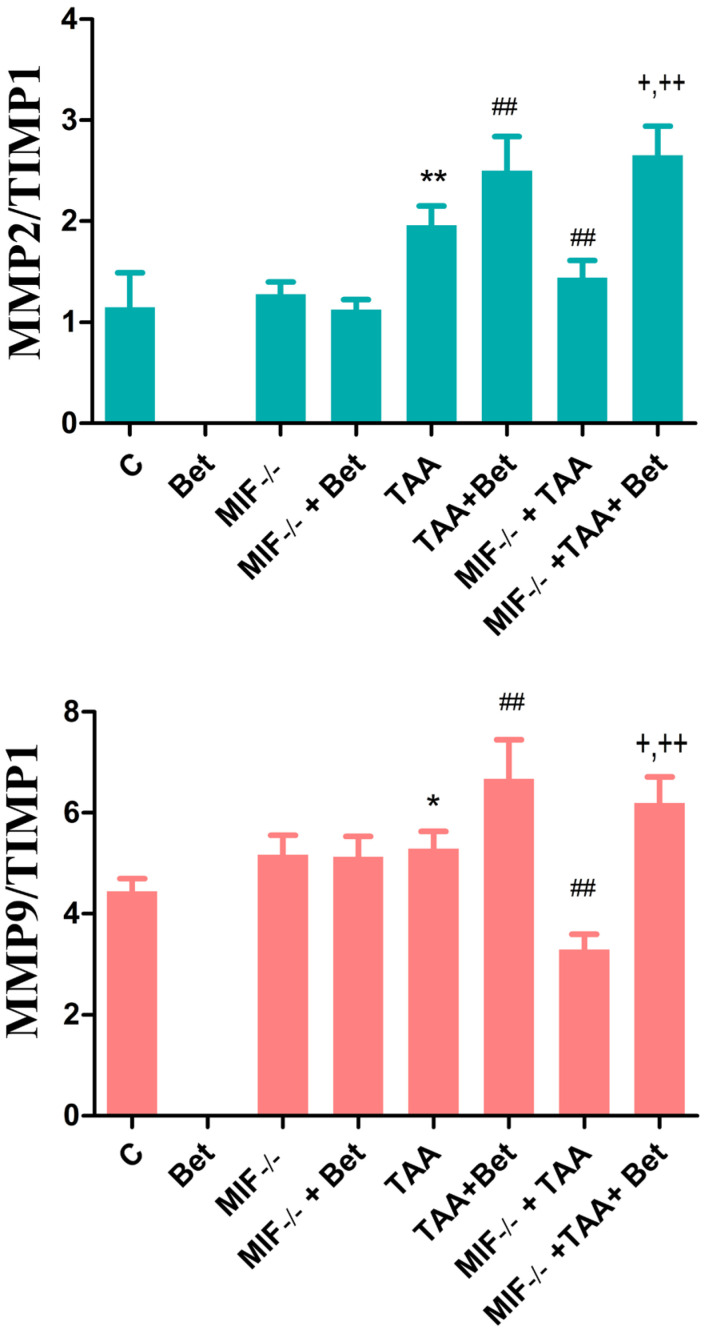
Effects of MIF and betaine on the changes in the MMP2/TIMP1 and MMP9/TIMP1 ratio in mice with liver fibrosis. The data are presented as mean ± SD from 10 mice per group. Significance of the difference was estimated by using one-way analysis of variance (ANOVA) with Tukey’s post hoc test; * *p* < 0.05; ** *p* < 0.01 vs. C; ^##^ *p* < 0.01 vs. TAA; ^†^ *p* < 0.05; ^††^
*p* < 0.01 vs. MIF−/−+TAA and TAA+Bet. For abbreviations, see Figure 1.

**Figure 4 biomedicines-12-01337-f004:**
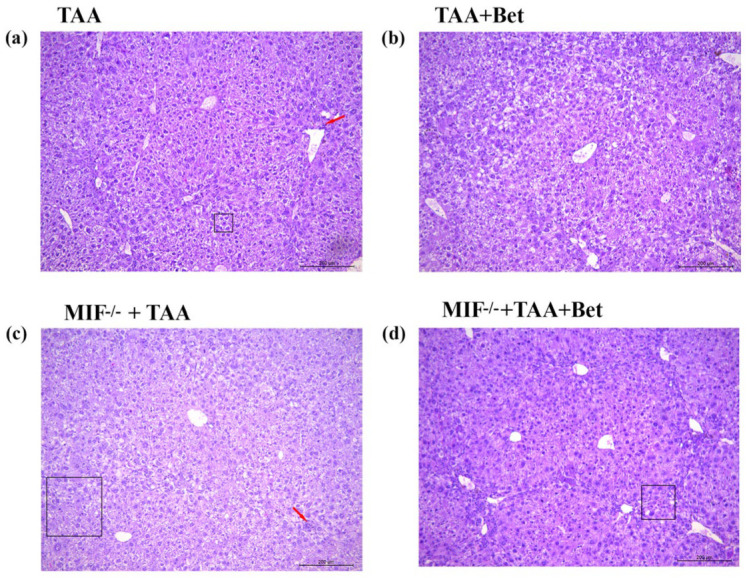
The effects of betaine and MIF on liver morphology in TAA-induced liver fibrosis in mice: TAA (**a**), TAA+Bet (**b**), MIF−/−+TAA (**c**) and MIF−/−+TAA+Bet (**d**) groups: Histological analysis by H&E (original magnification ×200). Micro- and macrovesicular fatty change (square); focal hepatocyte necrosis (red arrows). Pathohistological findings described in the text. For abbreviations, see Figure 1.

**Figure 5 biomedicines-12-01337-f005:**
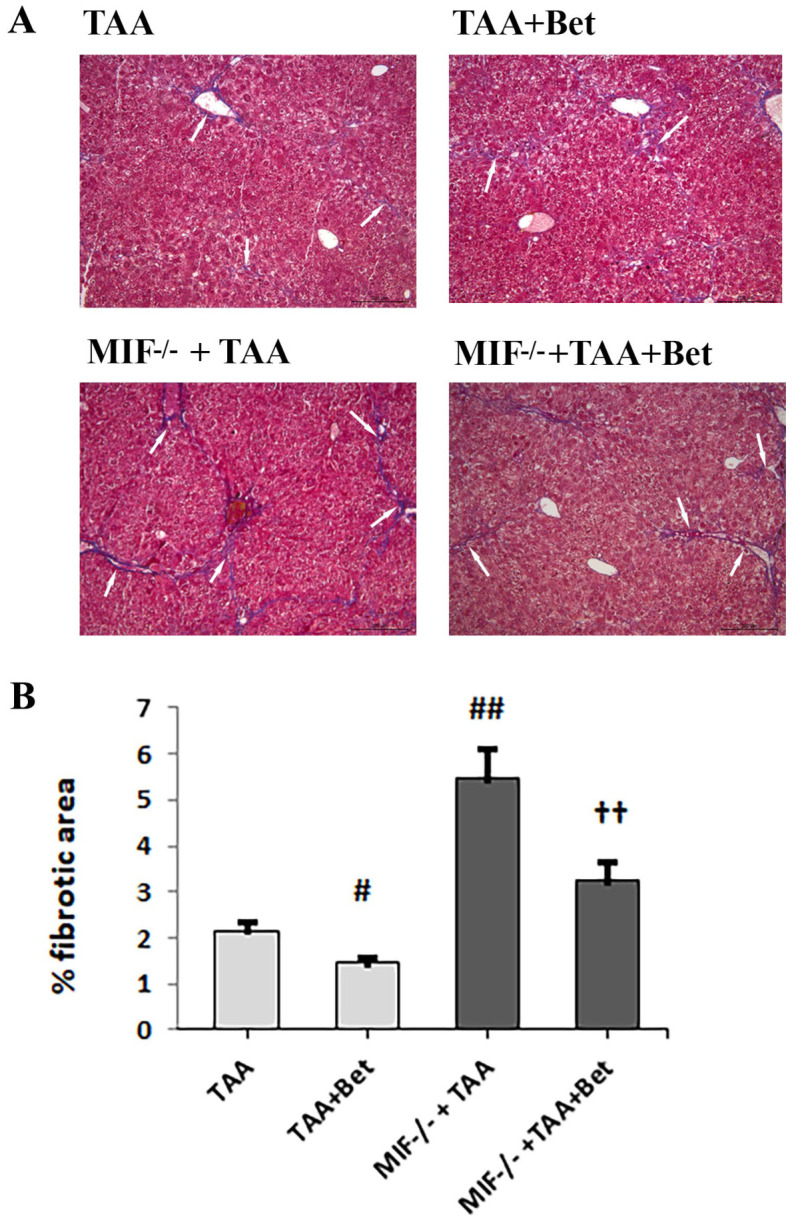
Histological findings in liver tissue (Masson’s trichrome staining, scale bar 200 μm, scoring systems with morphometric analysis of liver tissue fibrosis; quantification of percentage of fibrotic area). Effects of MIF and betaine supplementation fibrotic tissue accumulation in liver of mice with TAA-induced liver fibrosis (n = 10). White arrows show type I collagen (**A**). Quantification of fibrosis (**B**). Significance of the difference was estimated by two way analysis of variance (ANOVA) with Tukey’s post hoc test; ^#^
*p* < 0.05, ^##^
*p* < 0.01, vs. TAA; ^††^
*p* < 0.01 vs. MIF−/−+TAA and TAA+Bet. The data are presented as mean ± SD. For abbreviations, see Figure 1.

**Figure 6 biomedicines-12-01337-f006:**
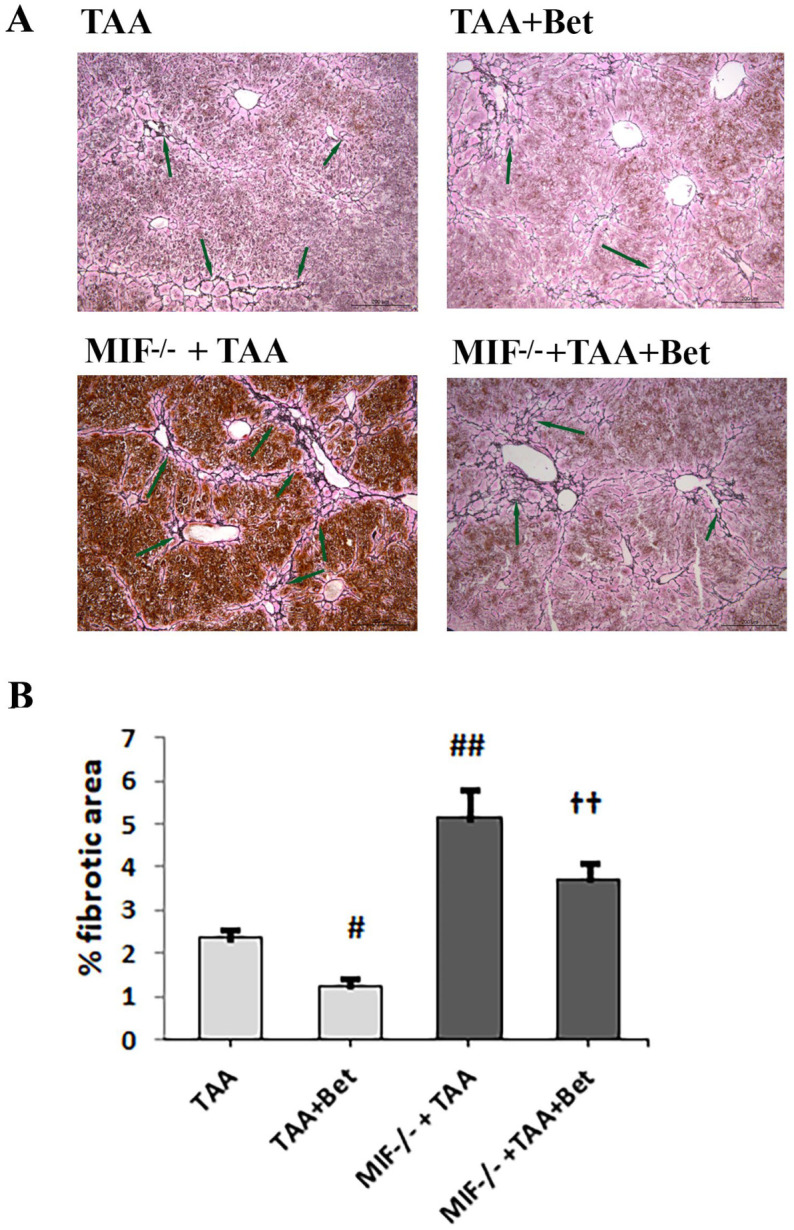
Histological findings in liver tissue (Reticulin staining, scale bar 200 μm, scoring systems with morphometric analysis of liver tissue fibrosis; quantification of percentage of fibrotic area). Effects of MIF and betaine on fibrotic tissue accumulation in liver of mice with TAA-induced liver fibrosis (n = 10). Green arrows show type III collagen (**A**). Quantification of fibrosis (**B**). Significance of the difference was estimated by two way analysis of variance (ANOVA) with Tukey’s post hoc test; ^#^
*p* < 0.05, ^##^
*p* < 0.01, vs. TAA; ^††^
*p* < 0.01 vs. MIF−/−+TAA and TAA+Bet. The data are presented as mean ± SD. For abbreviations, see Figure 1.

## Data Availability

Dataset available on request from the authors.

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
