# Peer review of "The Role of Macrophage Inhibitory Factor in TAA-Induced Liver Fibrosis in Mice: Modulatory Effects of Betaine"

_biomedicines, 2024, doi:10.3390/biomedicines12061337_

Round 1
Reviewer 1 Report
Comments and Suggestions for Authors
The paper presented by Radosavljevic and colligues clearly and thoroughly investigates The role of Macrophage Inhibitory Factor in TAA-induced Liver fibrosis in Mice: Modulatory effects of Betaine
In this study, both the rationale and the study on the animal model are clearly presented
The role of MMP2 and MMP9 as well as TIMP-1 are also investigated
Both the antifibrotic abilities of betaine are highlighted as well as the possibility of having pro-inflammatory effects. The work is well written
The graphs and figures are extremely clear and well explained.
Publication is recommended
Author Response
We thank You for the time you have taken to review our manuscript.
Thank You very much for the positive review of our manuscript.
Reviewer 2 Report
Comments and Suggestions for Authors
Dear Authors,
I have carefully reviewed your manuscript on the role of Macrophage Inhibitory Factor in TAA-induced liver fibrosis in mice. Overall, the study provides valuable insights into the potential therapeutic effects of Betaine in liver fibrosis. However, I have some suggestions for improvement:
1. In Figure 1A, it should be labeled as MIF-/- instead of MIF for clarity and accuracy.
2. The clarity of images in Figures 4 and 5 is relatively low. Please consider enhancing the resolution for better visualization.
3. Please ensure that the sample sizes are clearly indicated in the figure captions for transparency.
4. Sections 3.3 and 3.4 show a high degree of overlap in content, with some images being repeated. It may be beneficial to consolidate these sections and avoid redundancy.
5. It would be helpful to provide a brief introduction to MMP-2, MMP-9, TIMP-1, and other relevant factors in the Introduction section to enhance the comprehensibility of the article.
6. At the end of each subsection in the Results section, consider adding a brief summary to highlight the key findings.
7. Consider expanding the analysis to include more profibrotic mediators such as PDGF, VEGF, etc., to provide a more comprehensive understanding of the fibrotic process.
8. It may be beneficial to include in vitro cell experiments to complement the in vivo findings and strengthen the translational relevance of the study.
Overall, your study has significant potential, and addressing these points could further enhance the clarity and impact of your research. Thank you for your valuable contribution to the field of liver fibrosis research.
Best regards,
Comments on the Quality of English LanguageMinor editing of English language required
Author Response
Reviewer #2
We thank you for the time you have taken to review our manuscript and for your constructive comments and useful suggestions, which have greatly improved the quality of the manuscript and enabled us to improve it. We have thoroughly reviewed all the comments/suggestions. Our responses are provided below and are highlighted in the revised manuscript.
Comment 1: In Figure 1A, it should be labeled as MIF-/- instead of MIF for clarity and accuracy.
Reply 1: Thank you for drawing our attention to the error in Figure 1A. On the x-axis, MIF was changed to MIF-/- in all groups where necessary and is now clear and correct.
Comment 2: The clarity of images in Figures 4 and 5 is relatively low. Please consider enhancing the resolution for better visualization.
Reply 2: Thank you for your feedback. The highest resolution that we can provide is used in the manuscript. The photomicrographs are enlarged to make the changes more visible.
Comment 3. Please ensure that the sample sizes are clearly indicated in the figure captions for transparency.
Reply 3: Thank you for your comment. The sample sizes are indicated in all figure captions in the revised manuscript.
Comment 4: Sections 3.3 and 3.4 show a high degree of overlap in content, with some images being repeated. It may be beneficial to consolidate these sections and avoid redundancy.
Reply 4: We appreciate your comment. We have changed subsections 3.3. and 3.4. Following the suggestion of the Academic Editor, we have included the effects of MIF and betaine on the ratio of MMP2 and MMP9 over TIMP1 (Results section, subsection 3.3. The effects of MIF and betaine on the imbalance of MMP2/TIMP1 and MMP9/TIMP1 in the livers of mice with TAA-induced liver fibrosis). These results are also shown in Figure 3 of the revised manuscript. Furthermore, in order to avoid redundancy, we have converted Figure 3 to Figure 4 of the revised manuscript and removed all redundant content. Figure 4 of the revised manuscript contains photomicrographs showing hematoxylin and eosin staining of liver tissue (experimental groups: TAA, TAA+Bet, MIF-/-+TAA, and MIF-/-+TAA+Bet), and the description of the changes depicted in these photomicrographs can be found in the Results section, subsection 3.4. Effects of MIF and betaine on histology findings of liver morphology in mice with TAA-induced liver fibrosis, of the revised manuscript.
Comment 5: It would be helpful to provide a brief introduction to MMP-2, MMP-9, TIMP-1, and other relevant factors in the Introduction section to enhance the comprehensibility of the article.
Reply 5: Following the reviewer's suggestion, the Introduction section was enriched with the relevant information on matrix metalloproteinases, especially MMP2 and MMP9, and tissue inhibitors of metalloproteinases (Introduction section, 76-89, and 134-143).
Comment 6: At the end of each subsection in the Results section, consider adding a brief summary to highlight the key findings.
Reply 6: We appreciate your suggestion. A brief summary has been included in the Results section where we thought it would be useful.
Comment 7: Consider expanding the analysis to include more profibrotic mediators such as PDGF, VEGF, etc., to provide a more comprehensive understanding of the fibrotic process.
Reply 7: We thank you for your suggestion. Actually, PDGF has been already analysed in our manuscript (Results section, subsection 3.1. The effects of MIF and betaine on profibrogenic mediators (TGF-β1 and PDGF-BB) in TAA-induced liver fibrosis, Figure 1). Furthermore, TGF-β1 is a key mediator in the fibrotic response. In the experiments we are planning further, with the aim of deepening our research, we will definitely examine other profibrotic mediators among many.
Comment 8: It may be beneficial to include in vitro cell experiments to complement the in vivo findings and strengthen the translational relevance of the study.
Reply 8: We very much appreciate your suggestion. Our first endpoint was to investigate the role of macrophage inhibitory factor and the effects of betaine on liver fibrosis in animal model. Now that we have shown that betaine modulates the fibrogenic effect of MIF, we will definitely include in vitro experiments in our further research.
Reviewer 3 Report
Comments and Suggestions for Authors
English should be revised
The discussion should be expanded with current articles
The novelty of the article should be expressed more clearly in the introduction
By what method were damage scores made in tissues?
Author Response
Reviewer #3
We thank you for the time you have taken to review our manuscript and for your constructive comments and useful suggestions, which have greatly improved the quality of the manuscript and enabled us to improve it. We have thoroughly reviewed all the comments/suggestions. Our responses are provided below and are highlighted in the revised manuscript.
Comment 1: English should be revised.
Reply 1: The manuscript has been revised by our colleague who is a native English speaker and minor English corrections have been made.
Comment 2: The discussion should be expanded with current articles.
Reply 2: Thank you for your comment. We have included recent articles in the Discussion section, as well as recent articles considering ratio among matrix metalloproteinases, particularly MMP2 and MMP9, and tissue inhibitors of metalloproteinases. Please note that we have also considered the effects of MIF and betaine on the ratio of MMP2 and MMP9 to TIMP1 (Results section, subsection 3.3. The effects of MIF and betaine on the imbalance of MMP2/TIMP1 and MMP9/TIMP1 in the livers of mice with TAA-induced liver fibrosis; Figure 3).
Comment 3: The novelty of the article should be expressed more clearly in the introduction.
Reply 3: Following the reviewer's suggestion, the novelty of the article is emphasized at the end of the introduction section of the revised manuscript (Introduction sect. 76-89 and 134-143 ).
Comment 4: By what method were damage scores made in tissues?
Reply 4: Fibrosis was quantified by analyzing liver tissue samples stained with Masson’s trichrome and Reticulin with a digital image camera. The surface of the blue-stained area at a microscope magnification of x200 was calculated in 10 random fields on each section of each animal and presented as a percentage of the total liver cross-sectional area using the ImageJ software. Analysis was performed in accordance with Tissue pathways for liver biopsies for the
investigation of medical disease and focal lesion, The Royal College of Pathologists, October 2020 – October 2025, unique document number G064.
Round 2
Reviewer 2 Report
Comments and Suggestions for Authors
Dear Editor,
I have conducted a second review of the manuscript titled "Role of Macrophage Inhibitory Factor in TAA-Induced Liver Fibrosis in Mice: Modulatory Effects of Betaine" and I am pleased to report that all the issues I previously raised have been satisfactorily addressed by the authors. The key findings regarding the role of Macrophage Inhibitory Factor in liver fibrosis induced by TAA in mice have been clearly presented and supported by relevant data. The modulatory effects of Betaine on Macrophage Inhibitory Factor in the development of liver fibrosis have been well-explained, providing valuable insights into potential therapeutic strategies.
The authors have effectively revised the manuscript to enhance clarity and address the concerns raised during the first review. I recommend proceeding with the publication of this valuable research after the final checks are completed.
Thank you for the opportunity to review this manuscript, and I look forward to seeing it published in your journal.
Best regards,
Comments on the Quality of English LanguageIndividual handwriting needs to be adjusted
Reviewer 3 Report
Comments and Suggestions for Authors
The article can be accepted to this version.
Comments on the Quality of English LanguageGood